# Health-Related Content of TV and Radio Advertising of Dietary Supplements—Analysis of Legal Aspects after Introduction of Self-Regulation for Advertising of These Products in Poland

**DOI:** 10.3390/ijerph19138037

**Published:** 2022-06-30

**Authors:** Regina Ewa Wierzejska, Agnieszka Wiosetek-Reske, Magdalena Siuba-Strzelińska, Barbara Wojda

**Affiliations:** 1Department of Nutrition and Nutritional Value of Food, National Institute of Public Health NIH—National Research Institute, Chocimska St. 24, 00-791 Warsaw, Poland; bwojda@pzh.gov.pl; 2The Institute of Health Sciences, Stefan Batory State University, Batorego St. 64C, 96-100 Skierniewice, Poland; awiosetek@wp.pl; 3National Center of Nutritional Education, National Institute of Public Health NIH—National Research Institute, Chocimska St. 24, 00-791 Warsaw, Poland; mag.strzelinska@gmail.com

**Keywords:** dietary supplements, advertisements, consumer, health claims, legal regulations

## Abstract

Dietary supplements may have beneficial value but, by definition, they have no therapeutic effect. However, their labeling and especially the advertisements in the media, often make ungrounded health claims. The aim of the study was to analyze the content of audio–visual advertisements of dietary supplements for health and legal aspects in the context of the European Law and the 1 January 2020 Polish self-regulation between TV broadcasting companies and supplement manufacturers. Supplement advertisements broadcast across six TV and radio stations from 9–15 March 2020 were analyzed. Most of the analyzed advertisements complied with the legal requirements and included terms such as ‘supports’ or ‘facilitates’ body function, which are less definite in nature. Almost 30% of the advertised supplements made unproven claims on their effectiveness in various health situations, e.g., effective weight loss, thus assuring the addressee about the beneficial effect of dietary supplements in a given health context. Agreement on the rules and regulations governing supplement advertising resulted in a noticeable improvement in advertisement content, which will hopefully raise consumer awareness about the absence of therapeutic properties of dietary supplements.

## 1. Introduction

Legally, dietary supplements fall into the food category, even though both their form and dosage correspond to the drug category. The 2002 European Union (EU) Directive defined dietary supplements as foodstuffs which contain concentrated sources of nutrients or substances with a physiological effect, designated to supplement a typical diet [1]. However, the EU classification of dietary supplements lacks homogeneity due to inconclusive criteria. In some countries, preparations with identical content are registered as dietary supplements—based on food law, and others as OTC (over-the-counter) medicine—based on pharmaceutical law [2,3].

The legal status of any preparation is of vital importance in terms of product description on the package, the package insert and advertising. In the European Union, disease-related preventive and/or medicinal actions cannot be attributed (or even implied) to the mechanism of action of dietary supplements, as is the case for all foodstuffs [4]. In the United States, it is obligatory for supplement labeling to mention that the product is not meant to be used for diagnostic, therapeutic, or preventive purposes for any disease and its action is not verified by the Food and Drug Administration (FDA) [5,6]. Those requirements were stipulated to mark the demarcation line between foods and drugs in order to avoid consumer confusion.

The dietary supplement market is developing rapidly. The number of these products in the US market is estimated to exceed 90,000 products [7]. In Poland, the number of dietary supplements reported to the Chief Sanitary Inspector is increasing year-on-year. In 2014, producers or distributors reported their intention to place 4388 dietary supplements on the market, in 2016—7398, and in 2021 as many as 22,986 were notified [2,8]. The large-scale production of dietary supplements goes hand in hand with the interest of the public. In the United States, as many as 77% of adults declare supplement use [9], while in Poland 30–78% of adolescents and adults [10,11,12] and approximately 40% of children report supplement use [13]. The most frequently taken dietary supplements, both in Poland and worldwide, are vitamins and minerals, followed by omega-3 fatty acids [9,10,12,14]. Only one-quarter of supplement users take supplements recommended by their physician, which suggests that the overwhelming majority use dietary supplements of their own accord [5].

Rapid growth of dietary supplement markets, along with a great variety of supplement ingredients and purposes, is the reason why the manufacturers have started to deliberately bypass legal requirements and restrictions regarding the supplements, and to make various ungrounded claims, including health claims, on these products [7,15,16,17,18]. Importantly, Poland has noted an almost 20-fold increase in the number of health product advertisements, including dietary supplements, on the radio and TV in comparison to 1997. In 2015 alone, the staggering number of 1204 advertisements of dietary supplements appeared on TV in just four channels and approximately twice as many on the radio [19]. That means society is being ‘bombarded’ with advertisements which create the illusion that dietary supplements can solve any health problem. Moreover, pharmacists continue to report that the customers of their pharmacies mention particular advertisements when purchasing a product for health reasons [19]. The number of the literature reports about audio–visual advertisements of dietary supplements is limited. In Poland, as was demonstrated in the earlier study by Wierzejska (2014), dietary supplements in audio–visual advertisements were commonly presented as products which may eliminate various health complaints, using the image of a physician or even the sound of an ambulance [20]. In 2017, the highest number of complaints to the Advertising Ethics Committee, the ‘product and service’ subsection, referred to advertisements of dietary supplements [21]. The situation is even more serious when it comes to internet advertising. In some regions of Poland, the State Sanitary Inspectorate audit of internet websites revealed that objectionable content and presentation was detected in 12–90% of supplement advertisements between 2014–2016 [16]. Advertisements of supplements claiming protective action against coronavirus constitute the most recent example of such extremely inappropriate conduct [22,23].

In light of the above, it is hardly surprising that a considerable group of consumers fails to distinguish between dietary supplements and drugs, using the former as preventive or therapeutic measures for a number of health conditions [24,25,26,27]. At the same time, drug manufacturers attempt to draw consumer attention to the distinction between non-prescription medicines and dietary supplements. Advertisements of medicinal preparations include target phrases such as ‘*it is a drug, not a supplement*’, ‘*medicinal product*’, and ‘*therapeutic OTC product*’ [20].

Despite the fact that dietary supplements as a product category have been approved on the EU market relatively recently (20 years ago), time has quickly shown the need for legal changes and restrictions regarding market launch, labeling, and advertising areas [2,6,28,29]. In Poland, the 2017 Polish Supreme Audit Office revealed a distinct lack of consumer protection against those dietary supplements which are not compliant with legal requirements [16]. In the conclusion section, they stipulated that there exists a need for the Chief Sanitary Inspector and Chief Officer for Competition and Consumer Protection to monitor supplement advertising, especially in light of the deceptive advertising practices which portray dietary supplements as therapeutic products. As a result of the much heated debate about the need to introduce changes in legal regulations, TV broadcasting companies and manufacturers of dietary supplements signed a ‘broadcasting agreement about the rules and regulations for advertising dietary supplements’—i.e., self-regulation of dietary supplements in Poland [30]. The date of the agreement, thereafter, referred to as ‘agreement’, became effective on 1 January 2020. The document introduced a number of important changes, chief among them the rule that an advertisement cannot imply, in a manner which is not consistent with the law, an association between dietary supplement and improved health condition, or use the image of any medical professional. Additionally, every TV advertisement is obligated to include a following message in a graphic form: *‘dietary supplement; contains ingredients which support physiological functions of the body by supplementing a typical diet; has no medicinal properties’*. The information needs to cover no less than 10% of the screen and must be displayed for at least 5 s. However, it is worth mentioning that this agreement may postpone or stop urgent changes in the law regulating dietary supplements.

There is very little research in the literature that analyzes information about the properties of dietary supplements conveyed in advertisements of these products. In light of the steadily growing markets of dietary supplements and the role of advertising in the decision-making process to purchase a product, it seems feasible to evaluate the legal and self-regulatory compliance of supplement advertisements. Therefore, the aim of the study was to analyze the content of dietary supplement advertisements broadcast across TV and radio, by state and commercial parties, in terms of their health aspects.

## 2. Material and Methods

All advertisements of dietary supplements, broadcast in the course of 1 week, between 9 and 15 March 2020 (9 a.m. and 8 p.m.) by six radio and TV stations (Polskie Radio Program 1, Radio Zet, RMF FM, TVP 1, TVN, and Polsat) were analyzed. According to the 2019 report of the National Broadcasting Council, RMF FM and Radio Zet are the two most popular radio stations in Poland, whose audience predominantly consists of young people. In contrast, Polish Radio Program 1 was selected due to its great popularity among older populations [31] to avoid age bias. As far as TV is concerned, the most popular stations, with the largest audience base (according to the 2019 report) were selected [32]. The choice was based on the assumption that they attracted the highest number of advertisers. Radio and TV advertisements were recorded using a mobile phone and a digital video recorder, respectively, by various members of the team (6 people, in total), allowing for a repeated broadcasting of the material and a detailed analysis. If the same advertisement was broadcast in various stations on a number of occasions, in the analysis it was included only once. Only two dietary supplements were advertised both on TV and the radio; however, there was no difference in the health claims in advertising for these products. In the end, the number of the analyzed advertisements and the dietary supplements was the same.

Advertisements of dietary supplements which were presented as the announcements of program sponsors were excluded from the analysis due to the fact that the agreement conditions do not include product sponsoring and placement, and because such adverts did not include health claims, only the name of the producer and/or the dietary supplement. A complete text of the advertisement, including captions on the screen as well as the image message, was analyzed for compliance with the Polish Food Safety and Nutrition Act of 25 August 2006 [33], Regulation (EU) No. 1169/2011 of the European Parliament and of the Council of 25 October 2011 on the provision of food information to consumers [4], Regulation (EC) No. 1924/2006 of the European Parliament and of the Council of 20 December 2006 on nutrition and health claims made on foods [34], Commission Regulation (EU) No. 432/2012 of 16 May 2012, establishing a list of permitted health claims made on foods [35], as well as the 25 November 2019 broadcasting agreement on the rules and regulations governing dietary supplement advertising [30]. The analysis of the compliance of all advertisements from the present study was performed by one of the authors, who is an expert on dietary supplements and the related laws governing the labeling of foods and information for consumers.

The following practices were deemed non-compliant with the regulations:words and phrases which suggested that dietary supplements prevent or treat diseases or refer to such properties; or that were misleading as to the characteristics of the product and its nature; that attributed effects or properties which product does not possess; and that are unreliable (which is in breach of article 7 Regulation (EU) No. 1169/2011 of the European Parliament and of the Council as well as of the broadcasting agreement).health claims were not compliant with the legal registry of such claims (which is in breach of the Commission Regulation (EU) No. 432/2012), false information that guaranteed alleviation of health ailments, improved looks or mood; given that dietary supplement efficacy is not controlled by clinical studies, the producer may not promise that any effect on the consumer will be achieved (which is in breach of Regulation (EU) No. 1169/2011, Regulation (EC) No 1924/2006 and of the broadcasting agreement).

As far as the evaluation of the visual side of the TV advertisements was concerned, we scanned for the presence of representatives of medicine-related professions, targeting children, and checked whether the overall image was suggestive of a specific effect of the product in a specific clinical situation, and if it was in compliance with other agreement requirements.

## 3. Results

During the study period, 46 dietary supplements for different health conditions were advertised (41 and 5 for adults and children, respectively). Out of them, 27 (58.7%) supplements were advertised only on the radio, 17 (36.9%) only on TV, and 2 (4.4%) in both these media. The highest number of supplements (23.9%) targeted immune system-related topics (Table 1).

Overall, 76 different kinds of health claims, including 30 about the action of the product and 46 about the action of the included ingredients, were found in all advertisements of dietary supplements, not including claims using identical wording for advertising different products. Advertisements of 16 (34.8%) dietary supplements contained single claims, whereas the rest included two or more for each product. Advertisements of 10 (21.7%) dietary supplements assured the audience about the effectiveness of the given product, while advertisements of 2 (4.3%) guaranteed the effectiveness of individual ingredients of the preparation, both of which are not allowed. One advertisement of a dietary supplement (2.2%) contained both such claims. Additionally, the product was advertised for problems with urinary continence (i.e., urinary incontinence, which is recognized as a disease entity), and the content of the advertisement distinctly claimed that symptoms will cease if the product is used. Overall, out of 46 dietary supplements, 13 (28.2%) contained unreliable guaranteed action claims (14 claims), whose exact wording is presented in Table 2 (Part A and C). The content of the remaining advertisements was more undemonstrative, in the form of support claims of dietary supplement action (19 claims) (Table 2, Part B) and support claims action of supplement ingredients (43 claims) (Table 2, Part D and E). Additional information that ‘*the effect of the product action is the result of its component action*’ was included in the advertisements of four dietary supplements. The Polish version of the health claims are presented in Appendix A.

However, in the category of support claims related to supplement ingredients, two claims were not fully compliant with legal conditions. The first one, concerning: ‘*zinc has beneficial effect on tired eyes*’ (claim No. 1, Table 2, Part D) which—in that wording—has not been included on the list of health claims [35]. The other one pertained to extract of *Cola nitida* (claim No. 2, Table 2, Part D), where the extract was attributed the property of *‘intensive support*’ of fat burning and of helping ‘*effective reduction*’ of body weight in advertisements.

Other health claims regarding vitamins and minerals were compliant with the law. Notably, the presence of vitamin D in the supplement was the most frequent occurrence in the analyzed advertisements (17 dietary supplements; 36.9%). Furthermore, out of the 11 products which were recommended as immunity boosting, 7 mentioned vitamin D as their ingredient, whereas 6 out of 7 supplements which emphasized high nutrient content also mentioned vitamin D.

It is impossible to unequivocally verify the health claims regarding botanical as the registry of the allowable claims has not yet been determined. Although obviously all claims on a definite effect of action of any given supplement ingredient in the human body should not be used. The statement that ‘*extract of turmeric prevents accumulation of fats*’ and ‘*herbal extracts which eliminate the feeling of heaviness*’ is an example of such a claim (claims No. 1 and 2, Table 2, Part C).

The advertisements of all supplements contained information that the product in question was a dietary supplement. Furthermore, all advertisements of dietary supplements broadcast on TV contained the obligatory phrase ‘*dietary supplement, contains ingredients which support physiological function of the body, has no medicinal properties*’ introduced by the abovementioned agreement. The text was located at the bottom of the screen in a separate rectangle area, which occupied no less than 10% of the screen size (range: 12–14%), and was displayed for no less than 5 s (range: 5–9 s), which is compliant with the agreement conditions. Moreover, in compliance with the agreement, children were not the target of any of the analyzed advertisements and no physician or representative of medical profession listed in the agreement was portrayed in a TV commercial. Notably, an advertisement of one supplement was taped in a sports massage parlor and the supplement was recommended by professional masseur. As was stipulated in the agreement signed in Poland, no individual in an advertisement may be portrayed in a manner which suggests they are physiotherapists or that they recommend the use of the product, nor can any object present in the advertisement evoke associations with the rehabilitation process. Thus, the reliability of that particular advertisement needs to be questioned.

## 4. Discussion

In accordance with the Directive 2006/114/EC of the European Parliament and of the Council of 12 December 2006 concerning misleading and comparative advertising, ‘*misleading advertising’ means any advertising which in any way, including its presentation, deceives or is likely to deceive the persons to whom it is addressed or whom it reaches and which, by reason of its deceptive nature, is likely to affect their economic behaviour or which, for those reasons, injures or is likely to injure a competitor* [36]. The main role of food law is to protect the food consumers from misleading claims made by labeling, advertising, or other channels of communication (leaflet, internet, etc.) [4,37]. Dietary supplements belong to a specific food group but, as all foodstuffs, cannot be portrayed in a way which implies their therapeutic properties. Various literature reports often quote the ruling of the Court of Justice, which says that, to an average consumer, a product is portrayed as having therapeutic or preventive properties against a given disease not only if such a message is explicitly conveyed, but also if it is only implied [38].

In the analyzed advertisements, content such as ‘*effective weight loss*’, ‘*12 weeks and this is the effect*’, and ‘*problem with urinary continence—not my problem any more*’ was unreliable, and non-compliant with the regulations and conditions of the agreement because dietary supplements do not undergo clinical tests and manufacturers cannot be aware of the effects of their action. The message in TV advertisements related to dieting was additionally reinforced by images of women who rapidly lost weight, while those related to joints were linked with images of athletes, whose physical shape was supposedly the result of using the advertised product. On-screen information such as ‘*dietary supplements have no medicinal properties*’, displayed in a graphic form, softens the promise of their action, but it does not give the advertisers permission to disregard the law.

The message in most advertisements focused on the supporting actions in the body (e.g., *supports, facilitates, good for…*) and, as such, it should not create the impression that a given product guarantees proper function of the body or visibly improves health. However, further studies would be needed to determine the perception of such expressions among average consumers. When describing the nutrient content of dietary supplements, it is important to bear in mind that various claims made on supplement ingredients in an advertisement cannot at the same time relate to the products which contain these ingredients. Sweet bars with the following message on the front part of the package: ‘*milk pralines necessary for proper growth and development of children*’ are an example of improper practices on the Polish market. The ingredients (calcium and essential fatty acids) in fact have such properties but the information that the message applied to the ingredient action was placed at the back of the package.

Forty-six examples of claims related to ingredient action of the dietary supplements were found in the analyzed advertisements. Importantly, the vast majority of the claims in TV advertisements were presented on the TV screens as captions, using minuscule font sizes, which made the information practically invisible to the viewers. As far as claims related to the ingredients of dietary supplements are concerned, five cases of breach of food law were found in the present study. The claim ‘*active ingredients maintain proper muscle tone, including the muscles responsible for urinary continence*’ was the first example. Such health claim promises a certain action of the supplement ingredient, which cannot be clinically confirmed. The sentence ‘*zinc has beneficial effect on tired eyes*’ is another example of a controversial claim. A legally allowable claim would read: ‘*zinc contributes to the maintenance of normal vision**’.* Notably, the wording of the allowable claims may be modified by the manufacturers, on condition that they still make the same sense to the consumers, i.e., the advertised effect will not be exaggerated. The authors of the present study believe that claims related to tired eyes may make the consumers believe that zinc directly alleviates eye tiredness, which has no scientific grounds. All of the remaining health claims regarding vitamins and minerals were compliant with the law. Considerable popularity of vitamin D-related content in the analyzed advertisements is the result of the recent media reports about vitamin D deficits among the Polish population, which was confirmed by the literature [39,40].

Health claims associated with botanicals present a greater challenge. Until today, no list of health claims has been established and in 2012 the European Committee stopped the ongoing work of the European Food Safety Authority (EFSA) on the matter. Consequently, only a small number of the claims have been EFSA-evaluated, while most remain on the pending list. However, due to the formal complaints of food manufacturers presented to the Court of Justice of the European Union, the European Committee launched a project in 2016 whose results would help determine further course of action for the works on the abovementioned list for botanicals. Until that time, claims related to botanicals may be used by the manufacturers on condition they are not misleading and have scientific grounds [41].

However, that condition cannot be fulfilled in case of the claim that the ‘*extract of turmeric prevents accumulation of fats*’ and ‘*herbal extracts which eliminate the feeling of heaviness*’ which is another controversial claim found in the advertisements of dietary supplements and which assures the consumers about its beneficial action for the function of the body. The sentence, ‘*the e**xtract of Cola nitida intensively supports fat burning, helps effective reduction of body weight*’ is yet another claim, whose two fragments—‘*intensively supports*’ and ‘*effective reduction*’ are questionable as they are suggestive of very good weight-reducing properties, which is in violation of article 12 Regulation (EC) No 1924/2006 of the European Parliament and of the Council, stating that claims which make reference to the rate of weight loss shall not be allowed [34].

Among botanical-related health claims whose legal status remains unresolved, three examples related to garlic have been found in the analyzed advertisements: ‘*garlic supports natural defense mechanism of the body’, ‘garlic helps maintain healthy respiratory system’,* and *‘garlic supports anti-oxidative ability of the body’.* The first two claims have not been evaluated by the EFSA yet [42], while the third has received a negative review [43], so its use will most probably be legally banned in the future. Other examples of such claims include the following: *‘f**ield horsetail helps you keep healthy hair, skin and nails*’, which has also received a negative review [44], and *‘c**ayenne pepper helps reduce body weight*’, which is on the pending list [42].

An illegal use of the term ‘*safe*’ was found in one of the advertisements of dietary supplements due to the fact that no unsafe foods may ever be introduced onto the market [37]. Nevertheless, much improvement in that regard has been observed when compared to the previous study of the authors, when the word ‘*safe*’ was found to be commonly used in such context [20]. Additionally, in contrast with the previous study, no examples of usage of the phrase ‘*available without prescription*’ were found. First of all, such a phrase states the obvious, i.e., form of sales of a dietary supplement, and secondly it may blur the lines between dietary supplements and drugs, which is a violation of article 7 Regulation (EU) No. 1169/2011 of the European Parliament and of the Council, stating that an advertisement may not mislead a consumer regarding the nature of the product [4].

The amount of research that analyzes advertising of dietary supplements for compliance with the law is limited. According to a Polish study (Wierzejska), in 2014, out of 27 dietary supplements advertised in the course of one week on TV and the radio, the content of 12 (44.4%) advertisements was not compliant with the legal regulations for foods. Many of them assured the audience of quantifiable health benefits as a result of supplement use, some advertisements contained phrases such as ‘treats’ and ‘prevents’, and three advertisements used images/texts which suggested the individual in the advertisement was a physician [20]. The matter of advertising honesty in Poland has also been discussed in the 2015 report of the National Broadcasting Council [30]. The analysis of 32 advertisements of dietary supplements, broadcast in the course of one week in four different TV stations revealed that 28 (87.5%) of the advertisements failed to inform the viewers that the advertised product was in fact a dietary supplement, and some advertisements were filmed inside pharmacies, which might suggest the product in question was therapeutic. Bearing that in mind, the results of our study demonstrate an improvement in advertising honesty as far as dietary supplements are concerned, which is probably the result of both the self-regulation of their advertising in Poland and the debate on the need to introduce radical changes in the law.

In Spain, an analysis of radio advertisements of dietary supplements, broadcast in 2017, revealed that health care professionals and celebrities participated in 14% and 25% of these advertisements, respectively, which is an obvious breach of the law [17]. In that study, the authors did not analyze the compliance of the health claims with the legal regulations. In the United States, in their study evaluating press and TV commercials, Lee et al. found that in over 96% of the advertised supplements, the proposed claims had no scientific grounds, and over 17% included therapeutic claims [25]. Chung et al. demonstrated that over 84% of supplement advertisements in the press illegally mentioned particular diseases [45]. The use of treatment-related information for supplement labeling is an alarmingly common practice [46,47]. Crawford et al. reported that 11 out of the 12 investigated products included claims such as *‘clinically proven*’, while products claiming preventive or therapeutic actions against Alzheimer’s disease have become so common in the United States, that the FDA was forced to issue an official position, informing the public that no trials to confirm the effectiveness of dietary supplements were being conducted and that the products may in fact be dangerous, ineffective, and may cause delay in seeking medical help [48]. In a Japanese study that assessed the advertising of dietary supplements in the two largest national newspapers, it was found that after the spread of COVID-19, the number of advertisements attributing the effectiveness of supplements in infection prevention significantly increased [49]. However, the latest surveys revealed that the level of consumer trust decreases as far as dietary supplement advertisements are concerned [24,28,50], and that consumers expect proof which verifies the alleged effectiveness of these products, not to mention the fact that they support legislative initiatives which may ensure reliability and credibility of the information about supplements [28,51].

The study is not without limitations, chief among them is seasonal collection of the material (early spring) and the fact that the study period coincided with the coronavirus pandemic, which might have strongly influenced the broadcasting frequency of supplement commercials aimed to boost immunity. Therefore, the present study is selective as compared to the wide range of the available dietary supplements. The next limitation is the fact that advertisements were analyzed by only one expert on diet supplements and food law; therefore, the assessment has a rather subjective character. Another limitation is the lack of analysis of advertisement reception by an average consumer (listener, viewer). The food law often emphasizes that neither labeling nor advertising of a product may mislead a consumer [4,34,37]. Thus, the authors of the present study are aware the evaluation of an advertisement performed by the expert need not necessarily overlap with the opinion of the consumer. In light of the fact that dietary supplement sales in Poland continue to grow every year, it seems prudent to investigate the reception of their advertisements by an average consumer.

Regardless, the analysis of advert reliability as far as the alleged properties of dietary supplements are concerned may constitute a valuable source of information for bodies and individuals dealing with health education.

## 5. Conclusions

After the agreement between TV broadcasting companies and manufacturers of dietary supplements was signed and introduced, most of the analyzed advertisements have complied with the legal requirements. Almost 30% of the advertisements of dietary supplements made claims on beneficial guaranteed action of the preparation or its ingredients in a given health context which, in the absence of clinical trials of dietary supplements, is not allowable. At present, in the absence of legal regulations, it is not possible to unambiguously verify the compliance of claims related to botanical components. However, their effects are mostly presented in a way which is not definitive as far as action in a body is concerned. Therefore, it needs to be emphasized that as far as health claims are concerned, there has been noticeable improvement in the content of supplement advertisements, as compared to the previous years. It might change the attitudes of consumers towards the allegedly therapeutic effects of dietary supplements. Still, advertisement perception of an average consumer requires additional studies, especially in light of the fact that the recipients of TV advertisements tend to focus their attention on the images rather than the text on the screen. Until comprehensive changes in the legal regulations are introduced, which continues to be advocated by numerous public health experts in many countries, self-regulations aiming to ensure reliable advertisements of dietary supplements may help to protect the interest and health of consumers.

## Figures and Tables

**Table 1 ijerph-19-08037-t001:** List of the advertised dietary supplements.

Purpose	No. of Dietary Supplements	Trade Name	Active Ingredients
		Acti Vita-miner D_3_	Vitamins (E, B1, B2, B6, B12, C, D, biotin, niacin, pantothenic acid, folic acid), minerals (iron, zinc, iodine), *Tagetes erecta* extract
		Acti Vita-miner senior D_3_	Vitamins (E, B1, B2, B6, B12, C, D, biotin, niacin, pantothenic acid, folic acid), minerals (iron, zinc, iodine, selenium), *Tagetes erecta* extract
		Ascorvita Max	Vitamins (C, D), zinc
		Blu Kid	Vitamin C, zinc, *Sambucus nigra* juice and extract, *Citrus aurantium* extract
		Estabiom Baby	Vitamin D, *Lactobacillus rhamnosus*, *Bifidobacterium breve*, fructooligosaccharides
Immunity	11	Gripovita	Vitamin C, zinc, *Sambucus nigra* extract, *Hibiscus splendens* extract, *Ribes nigrum* dry concentrate, *Rubus idaeus* dry concentrate
		Molekin D_3_	Vitamin D
		Pelavo Multi	Vitamin C, honey, *Tilia cordata* extract, *Pelargonium sidoides* extract, *Rubus idaeus* juice
		Preventic D_3_	Vitamin D, shark liver oil, *Allium sativum* oil
		Rutinacea Junior	Vitamin C, rutin, *Ribes nigrum* concentrate, *Rosa canina* extract, *Rubus idaeus* juice
		Rutinacea Max D_3_	Vitamin C, minerals (zinc, selenium), rutozide, citrus bioflavonoids
		Atresan	Glucosamine, methylsulfonylmethane, *Zingiber officinale* extract, vitamin C, copper
		Collaflex	Vitamin C, collagen, chondroitin, hyaluronic acid, inulin
		D-Vitum	Vitamin D, *Linum usitatissimum* oil
Joints, bones	8	D-Vitum forte K2	Vitamins (D, K), *Linum usitatissimum* oil
		Molekin D_3_+K_2_	Vitamins (D, K)
		Molekin Osteo	Vitamins (D, K), calcium
		Vitrum 1250 Calcium	Vitamin D, calcium
		4Flex	Vitamin C, collagen
		Climea forte	Vitamins (D, E, B6, folic acid), calcium, *Glycine* Willd extract, *Humulus lupulus* extract, *Linum usitatissimum* powder
Menopause	3	Duo-Fem	Vitamins (C, D, B6, E), minerals (magnesium, zinc, chrome), *Trifolium pratense* extract, *Melissa officinalis* extract, *Humulus lupulus* extract
		Femitonina	Melatonin, *Glycine* Willd extract, *Trifolium pratense* extract, *Humulus lupulus* extract
Dieting	3	Liporedium	*Cola nitida* extract, *Capsicum annuum* extract, *Ilex paraguariensis* extract, *Garcinia cambogia* extract
		Slimbel-3 fazy	Vitamins (C, A, E, niacin, pantothenic acid, B1, B2, B6, folic acid, biotin, B12), minerals (magnesium, iron, manganese, copper, zinc), L-carnitin, *Sambucus nigra* juice, *Smilax officinalis* extract, *Citrus limon* juice, *Foeniculum vulgare* extract, *Rosmarinus officinalis* extract, Coenzyme Q10, *Fucus vesiculosus* extract, *Camellia sinensis* (green tea) extract, *Coffea arabica* extract, *Sambucus nigra* concentrate, *Ananas* Mill. concentrate, *Melissa officinalis* extract, *Tilia cordata* extract, *Sambucus nigra* extract, *Smilax officinalis* extract, *Verbena officinalis* extract
		Term Line	L-thyroxine, L-carnitin, chrome, citrus fruit extract, *Camellia sinensis* (green tea) extract, caffeine, *Paullinia cupana* extract, *Coffea arabica* extract, *Zingiber officinale* extract, *Capsicum annuum* extract, *Piper nigrum* extract
		Gold-Luteina	Vitamins (E, A), zinc, fish oil, lutein and zeaxantin from *Tagetes erecta*
Eyes	3	Nutrof Total	Vitamins (C, E, D), minerals (zinc, copper, selenium), lutein, zeaxantin, fish oil, *Crocus sativus* oil, *Vitis vinifera* extract
		Vizik max	Vitamins (C, B2, B6, B12, E, A), minerals (selenium, zinc), *Tagetes erecta* extract
		Apetizer Senior	Vitamins (B1, B2, B6, B12, folic acid, biotin, niacin, pantothenic acid), *Ribes nigrum* concentrate, *Pimpinella anisum* extract, *Cichorium intybus* extract, *Mentha piperita* extract, *Anethum graveolens* extract, *Citrus paradisi* extract
Appetite, digestion	3	Travisto	*Cynara scolymus* extract, *Curcuma longa* extract, *Mentha piperita* extract, *Anethum graveolens* extract
		Verdin Complexx	*Rosmarinus officinalis* powder, *Rosmarinus officinalis* extract, *Cynara scolymus powder*, *Cynara scolymus* extract, *Curcuma longa* extract, *Mentha piperita oil*
		Novanoc	Vitamin B6, melatonin, *Passiflora caerulea* extract, *Melissa officinalis* extract, *Eschscholzia californica* extract, *Arthrospira platensis* extract
Nerves, sleep	3	Positivum	*Melissa officinalis* extract, *Humulus lupulus* extract, *Crocus sativus* extract
		Valerin Sen	Vitamin B6, magnesium, *Melissa officinalis* extract, *Humulus lupulus* extract, L-theanine, L-tryptophan, Gamma-aminobutyric acid (GABA), *Crocus sativus* extract
Hair	2	Skrzypolen	Vitamin (D, A, E, biotin), minerals (selenium, zinc), taurine, L-methionine, collagen, *Equisetum arvense* extract, *Linum usitatissimum extract*, *Urtica dioica* extract
		Skrzypovita	Vitamins (C, B1, B2, B6, A, E, biotin, pantothenic acid), minerals (zinc, copper), para-aminobenzoic acid (PABA), collagen, hyaluronic acid, *Equisetum arvense* extract, *Urtica dioica* extract
Liver	1	Hepaslimin	L-ornithine, choline, *Ilex paraguariensis* extract, *Cynara scolymus* extract, *Curcuma longa* extract, *Cichorium intybus* extract
Memory	1	Bilomag Plus	Vitamins (B6, pantothenic acid), magnesium, *Ginkgo biloba* extract, lecithin
Heart	1	Gold Omega 3 D_3_ + K_2_	Vitamins (D, K, E), fish oil
Urinary incontinence	1	Feminost	Vitamins (D, B12), *Cucurbita pepo* extract, *Vaccinium oxycoccos* juice, *Glycine clandestina* extract, *Urtica dioica* extract
Throat	1	Fiorda	Lactoferrin, *Cetraria islandica* extract, *Rosa canina* extract, *Rubus ideaus* concentrate, *Sambucus nigra* juice
Pregnancy	1	Profolium	5-methyltetrahydrofolic acid
General support for the normal functioning of the body	4	Vitotal	Vitamins (C, E, A, B1, B2, B6, B12, D, K, niacin, pantothenic acid, biotin), minerals (zinc, iron, magnesium, manganese, copper, iodine, molybdenum, chrome, selenium), collagen, L-cysteine, L-methionine, lycopene, lutein, *Equisetum arvense* extract, *Vitis vinifera* extract
		Solevitum D3	Vitamin D
		NeoMag forte D3	Vitamins (D, B6), magnesium
		D-Vitum DHA	Vitamin D, *Schizochytrium* sp. oil

**Table 2 ijerph-19-08037-t002:** Action of the advertised supplements or their ingredients.

ACTION OF DIETARY SUPPLEMENTS ACCORDING TO ADVERTISERS
**Type of Claim**
**Part A.** **Unreliable guaranteed action claims** (non-compliant with Regulation (EU) No. 1169/2011, Regulation (EC) No. 1924/2006 and with the agreement)
**1.** *12 weeks and this is the effect (images of sporty people, indicating a well-functioning osteoarticular system)* **2.** *I use (name of the product) …and I no longer fear hot flashes, mood swings, night sweats and excessive* *weight* **3.** *(name of the product) … and I remember everything. I use it so my memory will never fail me* **4.** *The kilograms disappear forever, effective weight loss* **5.** *For losing weight—verified and effective* **6.** *With … (name of the product) I no longer worry about cellulite* **7.** *You will calm down, you will get a good night’s sleep* **8.** *(name of the product) … means healthy liver and additionally healthy weight* **9.** *(name of the product) … helped me; problem with urinary continence—not my problem any more* **10.** *(name of the product) … equals a good night’s sleep* **11.** *Your hair becomes thicker, more shiny and healthier*
**Part B.** **Support claims**
**1.** *Supports immunity* **2.** *A way to boost your immunity* **3.** *Allows you to take care of your immunity* **4.** *F* *or problems with appetite,* *promotes appetite and facilitates digestion* **5.** *Aids digestion and reduction of excess gasses, supports stomach, liver and gut function* **6.** *Supports collagen production* **7.** *Good for your joints* **8.** *For strong and healthy bones* **9.** *Protects your throat* **10.** *Takes care of the microbiome* **11.** *Relieves menopausal symptoms* **12.** *Complete supports your sight* **13.** *Supports proper vision* **14.** *Helps me keep my weight* **15.** *Helps you fall asleep faster* **16.** *Supports uninterrupted sleep and gives you energy* **17.** *Helps deal with stress* **18.** *A way to lose weight* **19.** *Offers complex support*
**ACTION OF SUPPLEMENT INGREDIENTS ACCORDING TO ADVERTISERS**
**Type of Claim**
**Part C.** **Unreliable guaranteed action claims** (non-compliant with Regulation (EU) No. 1169/2011, Regulation (EC) No. 1924/2006 and with the agreement)
**1.** *Extract of turmeric prevents accumulation of fats* **2.** *Herbal extracts for improved digestion, which eliminate the feeling of heaviness* **3.** *Active ingredients maintain proper muscle tone, including the muscles responsible for urinary continence*
**Part D.** **Unreliable support claims**
**1.***Zinc has beneficial effect on tired eyes *(non-compliant with the Commission Regulation (EU) No. 432/2012) **2.***Extract of Cola nitida intensively supports fat burning, helps effective reduction of body weight* (non-compliant with Regulation (EC) No. 1924/2006)
**Part E.** **Other support Claims**
**1.** *Vitamin D contributes to normal function of the immune system* **2.** *Vitamin D contributes to maintenance of normal bones, teeth, and muscle function* **3.** *Vitamin D is essential for healthy bones* **4.** *Vitamin D contributes to normal absorption of calcium* **5.** *Vitamin D* *contributes to healthy growth/development of children * **6.** *Vitamin D is vital for immunity* **7.** *Vitamin D supports immunity* **8.** *Vitamin C contributes to normal collagen formation for the normal function of skin and cartilage* **9.** *Vitamin C contributes to the normal function of the immune system* **10.** *Vitamin A contributes to the normal function of the immune system* **11.** *Vitamin A contributes to the maintenance of normal skin* **12.** *Vitamin A supports immunity* **13.** *Vitamin K_2_ for healthy bones* **14.** *Biotin contributes to the maintenance of normal hair and skin* **15.** *Biotin contributes to normal psychological function* **16.** *Zinc contributes to the maintenance of normal vision* **17.** *Magnesium contributes to normal functioning of the nervous system* **18.** *Calcium contributes to normal bone, teeth, and muscle function* **19.** *Omega-3 fatty acids contribute to normal cardiac function* **20.** *DHA contributes to normal brain function in children* **21.** *Melatonin helps you fall asleep* **22.** *Garlic supports anti-oxidative ability of the body and its natural defense mechanism* **23.** *Garlic supports defense mechanisms and helps maintain healthy respiratory system * **24.** *Extract of rosa canina supports immunity * **25.** *Extract of peppermint helps digestion and proper function of the digestive tract, and maintain healthy* *stomach* **26.** *Extract of clover helps alleviate menopausal symptoms* **27.** *Extract of humulus strobiles helps alleviate menopausal symptoms* **28.** *Extract of humulus strobiles contributes to peaceful sleep* **29.** *Cayenne pepper helps reduce body weight* **30.** *Yerba mate helps maintain proper body weight* **31.** *Extract of artichoke leaves supports detoxication, stimulates digestive acid secretion, and helps maintain* *healthy liver; assures intestine comfort* **32.** *Extract of dill flowers supports digestion, elimination of excess gas, and helps proper digestion of fats/lipids* **33.** *Extract of turmeric helps you maintain normal liver function* **34.** *Extract of saffron crocus increases emotional balance* **35.** *Extract of grape seeds* *helps maintain proper body weight and reduce cellulite* **36.** *Artichoke, rosemary, turmeric help digestion and digestive comfort* **37.** *Field horsetail helps you keep healthy hair, skin, and nails* **38.** *Flax supports weight control* **39.** *Amino acids are the building blocks of keratin, naturally found in hair and eyelashes* **40.** *Ingredients help maintain concentration and mental agility* **41.** *Ingredients soothe hot flashes and irritability, promote peaceful sleep and youthful appearance*

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
