# Peer review of "Health-Related Content of TV and Radio Advertising of Dietary Supplements—Analysis of Legal Aspects after Introduction of Self-Regulation for Advertising of These Products in Poland"

_ijerph, 2022, doi:10.3390/ijerph19138037_

Round 1

Reviewer 1 Report

This is a very descriptive paper. We have no insights into how (potential) consumers perceive the messages. That is fundamentally important.

Discussion of vitamins, which by definition are necessary for existence and do have a body of research particularly around Vitamin D and Covid, in terms of being unproven is problematic.

There is a well established pattern of "voluntary agreements" in advertising (in general) in advertising specific products around the world to avoid legislation and reduce impact on the industry who are voluntarily agreeing. That literature is missing from the paper and might provide a basis for more interesting implications and connection to a wider literature.

What should change as a result of this research?

Author Response

Thankful for your time spent on the review below we present our explanations to the questions asked.

This is a very descriptive paper. We have no insights into how (potential) consumers perceive the messages. That is fundamentally important.

The authors realize that research on how the advertising of dietary supplements is perceived by consumers/patients is very relevant. Therefore, we intend to conduct such a study in the future, but this particular study is about how nutrition experts evaluate supplements’ advertisements.

Discussion of vitamins, which by definition are necessary for existence and do have a body of research particularly around Vitamin D and Covid, in terms of being unproven is problematic.

There is no doubt that vitamins are essential for the body, and nowhere in the article do the authors contradict this. Our task was to analyze whether the messages in the advertising of dietary supplements do not promise too great impact on health, and it is expected that many potential consumers do not have vitamin deficiencies in the body. The literature shows that when it comes to nutrients, the principle "the more the better" is not appropriate.

There is a well established pattern of "voluntary agreements" in advertising (in general) in advertising specific products around the world to avoid legislation and reduce impact on the industry who are voluntarily agreeing. That literature is missing from the paper and might provide a basis for more interesting implications and connection to a wider literature.

In accordance with the Directive 2006/114/EC of the European Parliament and of the Council of 12 December 2006 concerning misleading and comparative advertising  ‘misleading advertising’ means any advertising which in any way, including its presentation, deceives or is likely to deceive the persons to whom it is addressed or whom it reaches and which, by reason of its deceptive nature, is likely to affect their economic behaviour or which, for those reasons, injures or is likely to injure a competitor.

This information was added to the article. The authors of this study are scientists in the field of nutrition and have made the analysis of advertisements based on the regulations of the food law. 

What should change as a result of this research?

It is obvious in science that one study will not solve the problem right away. However, the results of the study may constitute a valuable source of practical knowledge for public health authorities and be furnish an argument in the ongoing discussion on dietary supplements in recent years. Currently, there is hardly any such research in the literature.

We hope that our explanations will be sufficient and will meet your understanding.

Reviewer 2 Report

The article is very interesting and indicates the expert knowledge of the authors. I have only minor suggestions on how to improve it.

Introduction

The introduction is well written. The only thing that should be done is to clarify the aim of the study a bit. It can be done by adding research questions.

Material and Methods

There is a problem with the lines 152-164 (layout).

It is not clear from this section what kind of analysis was done (content)?

How many advertisements were broadcast at all? How many of them were advertisements for OTC? How many of them were advertisements for dietary supplements?

Results

Please mind that Part E has the title Other support claims, and line 204 is starting “Other health claims regarding”.

Author Response

Thankful for your time spent on the review below we present our explanations to the questions asked.

The article is very interesting and indicates the expert knowledge of the authors. I have only minor suggestions on how to improve it.

Introduction

The introduction is well written. The only thing that should be done is to clarify the aim of the study a bit. It can be done by adding research questions.

There is very little research in the literature that analyzes information about the properties of dietary supplements conveyed in advertisements of these products. This sentence was added to the article.  The authors believe that the aim of the study is quite precisely formulated.

Material and Methods

There is a problem with the lines 152-164 (layout).

It is not clear from this section what kind of analysis was done (content)?

How many advertisements were broadcast at all? How many of them were advertisements for OTC? How many of them were advertisements for dietary supplements?

As indicated in lines 140-147, the contents of the advertisements was screened in terms of their compliance with the food law. However, it should be taken into account that the legal requirements in particular regulations are very similar, e.g. according to the Regulation (EU) No. 1169/2011 of the European Parliament and of the Council food information shall not be misleading, particularly as to the characteristics of the product, its nature and properties, by attributing to the product effects or properties which it does not possess; shall not attribute to any food the property of preventing, treating or curing a human disease, nor refer to such properties, while in accordance with the Regulation (EC) No 1924/2006 nutrition and health claims shall not be false, ambiguous or misleading; shall be based on and substantiated by generally accepted scientific data.

Therefore, all messages in advertisements were, on the one hand, subjected to comprehensive review, and on the other hand, checked in terms of compliance with each legal provision.

The Authors have not counted the general number of advertisements. The same ads were repeatedly broadcast in the same day and throughout the week, nevertheless the number of ads was not the subject of this study. According to our estimation, in a course of one day approximately 40 dietary supplements ads were broadcast in a single radio station. However, this kind of data cannot be reliably presented in the text of the article.  We were not interested in the total number of advertisements for all products and services, but only in dietary supplements advertisements. Having recorded individual "advertising blocks," we were generally deleting advertisements of other products, such as washing powder, cosmetics, etc. This is why we do not have information on the total number of advertisements on radio and television stations.

We did not need this kind of data because the aim of the study was not to establish the number of advertisements, or not even the frequency of advertising for a particular supplement, but to obtain the number of dietary supplements advertised.

Dietary supplements advertisements (including recurrent advertisements for the same supplements) were broadcast repeatedly. In radio programs, advertising blocks were on air almost at fixed times, typical for a given radio station, always twice an hour, each block lasting about 4 minutes. For example, RMF FM broadcast commercials at 9.20 and 9.45, Polskie Radio Pr. 1 at 9.05 and 9.55, and the same pattern was followed in subsequent hours of their airtime. We estimate that there were about six advertisements of various products in each advertising block broadcast on the radio, of which two would generally promote dietary supplements. Therefore, each day in the course of 11 hours covered by the study (9 am - 8 pm), 22 advertising blocks were broadcast on each radio station, which roughly included about 44 dietary supplements advertisements (about 33% of all advertisements). As far as television is concerned, the digital video footage of the Polsat commercial TV station preserved to-date (11 March 2020, 15.00 - 20.00) shows that at that time, there were 12 advertising blocks, each built of about nine advertisements, of which dietary supplement advertisements appeared in half of those blocks (accounted for around 5% of all the advertisements). On public television (TVP 1) channel, supplements ads appeared more frequently and may have accounted for about 15% of all advertisements. On public television, unlike in commercial television, programs are not interrupted by advertisements.

At the same time, we would like to mention that in Poland, according to Article 16(3) of the Broadcasting Act of 29 December 1992 (as amended), advertisements and teleshopping must not take up more than 12 minutes per clock hour. Assuming an average duration of one commercial of approximately 30 seconds, according to our estimates, this results in approximately 11 minutes per hour.  

Results

Please mind that Part E has the title Other support claims, and line 204 is starting “Other health claims regarding”.

In part E there are all health claims presented regarding ingredients of dietary supplements other than unreliable guaranteed action claims (part C) or unreliable support claims (part D). The study generally looked at all health claims contained in advertising.

In accordance with the Regulation (EC) No. 1924/2006 of the European Parliament and of the Council of 20 December 2006 on nutrition and health claims made on foods ‘health claim’ means any claim that states, suggests or implies that a relationship exists between a food category, a food or
one of its constituents and health.

We hope that our explanations will be sufficient and will meet your understanding.

Reviewer 3 Report

The article is overall well written and the topic is relevant both for the legislators, consumer associations and diet supplements manufactures. I have only a comment concerning the scope of the analysis. Since the authors are analyzing advertisements they could enrich the analysis by introducing a theoretical background guided by the so-called Frame Theory.

This approach focuses on the process by which advertisements’ makers “select some aspects of a perceived reality [of supplements’ chemical functions] and make them more salient in the communicating context, in such a way as to promote a particular problem definition, causal interpretation, moral evaluation, etc.” (Entman, 1993).

By introducing this theoretical frame (and the corresponding, “easy” statistical analysis) the article wold reach a wider audience. Besides, results would be more relevant for the scientific contribution becoming something more than a mere (descriptive) statistical report.

Minor:

Please check the wording:

 Line 199 (double use of not)

Bibliography:

Entman, R. M. 1993. “Framing: Toward Clarification of a Fractured Paradigm.”
Journal of Communication 43 (4): 51–8. 

Author Response

Thankful for your time spent on the review below we present our explanations to the questions asked.

The article is overall well written and the topic is relevant both for the legislators, consumer associations and diet supplements manufactures. I have only a comment concerning the scope of the analysis. Since the authors are analyzing advertisements they could enrich the analysis by introducing a theoretical background guided by the so-called Frame Theory.

This approach focuses on the process by which advertisements’ makers “select some aspects of a perceived reality [of supplements’ chemical functions] and make them more salient in the communicating context, in such a way as to promote a particular problem definition, causal interpretation, moral evaluation, etc.” (Entman, 1993).

By introducing this theoretical frame (and the corresponding, “easy” statistical analysis) the article wold reach a wider audience. Besides, results would be more relevant for the scientific contribution becoming something more than a mere (descriptive) statistical report.

Bibliography:

Entman, R. M. 1993. “Framing: Toward Clarification of a Fractured Paradigm.” Journal of Communication 43 (4): 51–8. 

 The authors appreciate signaling the existence of Frame Theory as a platform that can be used in social communication in the media. However, our goal was only to analyze the content of the advertisements as to the health effects of supplements, bearing in mind the conditions of food law.

The authors do not consider themselves experts in the “philosophy” of advertising production, and therefore it would be inappropriate for us to extend the article upon the issues of advertising concepts adopted by producers, and their effectiveness.

We hope that our explanations will be sufficient and will meet your understanding.

Minor:

Please check the wording:

 Line 199 (double use of not)

 Thank you very much for your perceptiveness. The error has been corrected

Round 2

Reviewer 1 Report

NA